# Digital Medical X-ray Imaging, CAD in Lung Cancer and Radiomics in Colorectal Cancer: Past, Present and Future

**Jacobo Porto-Álvarez** [1,*], **Gary T. Barnes** [2], **Alex Villanueva** [1], **Roberto García-Figueiras** [1], **Sandra Baleato-González** [1,*], **Emilio Huelga Zapico** [1] and **Miguel Souto-Bayarri** [1]

1   Department of Radiology, Complexo Hospitalario Universitario de Santiago de Compostela, 15706 Santiago de Compostela, Spain
2   Department of Radiology, University of Alabama Birmingham, Birmingham, AL 35294, USA
*   Correspondence: jacobo.porto.alvarez2@sergas.es (J.P.-Á.); sandra.baleato.gonzalez@sergas.es (S.B.-G.)

**Abstract:** Computed tomography (CT) introduced medicine to digital imaging. This occurred in the early 1970s and it was the start of the digital medical imaging revolution. The resulting changes and improvements in health care associated with digital imaging have been marked, are occurring now, and are likely to continue into the future. Before CT, medical images were acquired, stored, and displayed in analog form (i.e., on film). Now essentially all medical images are acquired and stored digitally. When they are not viewed by computer, they are converted to an analog image to be seen. The application of computer algorithms and the processing of digital medical images improves the visualization of diagnostically important details and aids diagnosis by extracting significant quantitative information. Examples of this can be seen with CAD and radiomics applications in the diagnosis of lung and colorectal cancer, respectively. The objectives of this article are to point out the key aspects of the digital medical imaging revolution, to review its current status, to discuss its clinical translation in two major areas: lung and colorectal cancer, and to provide future directions and challenges of these techniques.

**Keywords:** computer-aided diagnosis (CAD); artificial intelligence (AI); radiomics; radiogenomics; machine learning; deep learning; colorectal cancer; lung cancer; tumor mutations





## 1. Introduction

Computed tomography (CT) introduced medicine to digital imaging. This occurred in the early 1970s and was the start of the digital medical imaging revolution. The resulting changes and improvements in health care associated with digital imaging have been marked, are occurring now, and are likely to continue in the future. Before CT, medical images were acquired, stored, and displayed in analog form (i.e., on film). Now essentially all medical images are acquired and stored digitally. When they are not viewed by computer, they are converted to an analog image for the human eye to be seen. The application of computer algorithms and the processing of digital medical images allows for the enhancement of details of diagnostic importance and the extraction of significant quantitative information for diagnosis. It also allows focusing on aspects of the image that suggest pathology, i.e., computer-aided diagnosis (CAD), or extracting and analyzing quantitative information which facilitates a predictive model (radiomics) but also generates large databases with this information, from which hypotheses can be generated through its analysis ("data mining"). This article will review the key aspects of the digitization of medical imaging focused on CAD and deep learning-based computer-aided diagnosis (DL-CAD), which are examples of computer vision applied to assist medical imaging diagnosis. It will also discuss the clinical translation of CAD and radiomics in lung and colorectal cancer, respectively, and will evaluate future trends in these technologies.

## 2. Past

### 2.1. Digitization

In the 1960s, digital computers began to migrate slowly into medical imaging, but the transforming event was the introduction of computed tomography (CT) into medical imaging in the early 1970s. In CT, mathematical algorithms created images from many X-ray measurements across multiple projections. Apart from imaging acquisition and formation, digital imaging also opened the possibility for multiple advances including image processing and improvement (i.e., noise suppression), multiplanar reconstruction, three-dimensional (3D) imaging, and transmission to remote places accompanied by the pertinent information of the patients [1]. Nowadays, there are three potentially transformative innovations. One is in the field of archiving and communications, where there will be interesting developments related to storage in the picture archiving communication systems (PACS) and also to the speed of transmission and recovery of images, as well as technologies at lower costs. The second is the increasing interconnection of all types of devices and objects through the network. Medical reports can be written in support (tablet, telephone, etc.) that will not necessarily have to be located in a hospital or other health center. A third group of innovations will be related to technologies that may improve diagnosis and treatment, either by lowering the price of the medical services themselves, helping people with disabilities, improving the quality of life of the elderly, or developing algorithms for computer-aided diagnosis and quantitative extraction of imaging features from medical scans ("radiomics") (Figure 1). Digitization strategy in the health environment rests on two strengths: the logical integration of the image data (PACS) with the patient's clinical and demographic data and the transmission of data and images from one location to another [2].

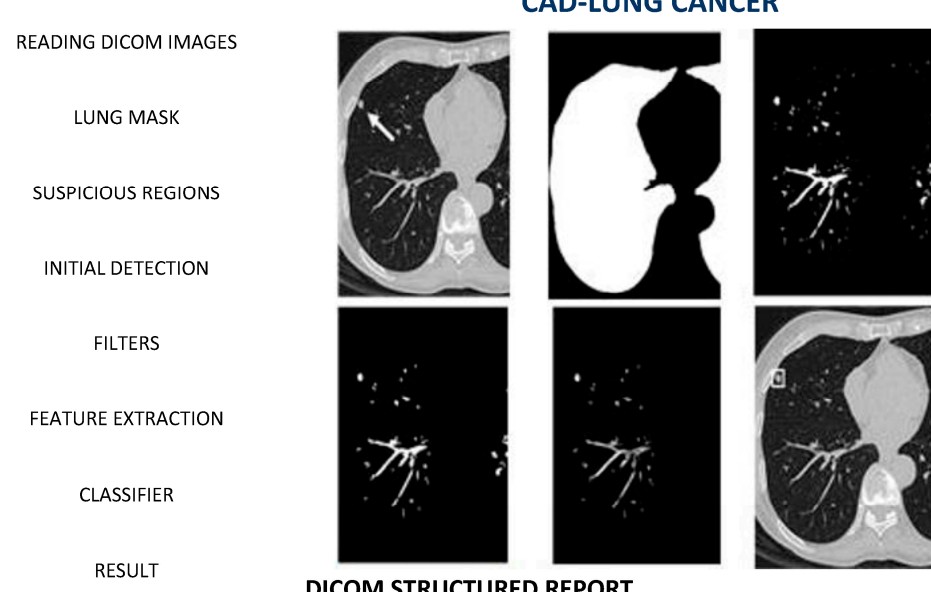

**Figure 1.** Standard scheme in CAD: From Acquisition to Data analysis [3].

### 2.2. From Analogic to Digital

Forty years ago, at the University of Alabama Birmingham, a team of medical physicists and radiologists led by Gary Barnes and Robert Fraser worked on a prototype for Digital Chest Radiology [4,5]. They subsequently published an article with other leaders in the field which prophesized the "fully digital radiology department of the future" [6]. Since then, these techniques have colonized much of the territory of medicine and represent key tools in medical imaging. Now, as we have pointed out, the digital age is amplified by a

phenomenon of a different nature: artificial intelligence. Its application in digital medicine seeks that computers help doctors in several ways:

(1) Finding signs of disease on diagnostic images.
(2) Helping in the operating room, locally or remotely, without tiredness or tremors.
(3) Combining patient information in a way that is useful for diagnosis and research. In other words: big data, for predictive analysis of large amounts of information.

These daring proposals were stated, among others, by Kunio Doi, from the University of Chicago (USA) [7]. Moreover, they were, like almost all revolutionary ideas, received with skepticism. Furthermore, its consequences were impossible to glimpse. However, at the Annual Meeting of the Radiological Society of North America in Chicago, IL, USA, we encountered careful staging by his group, it was easy to see that things were serious.

CAD has gone through several stages. There have also been "delays", a large part of which have had to do with the fact that the research drive of the first period has had no one to articulate it from the business perspective. It is not just that there have been technical implementation issues. It is difficult to combine two such different logics, neither of which is in a position to replace the other. On the one hand, human creativity speculates, tries, fails and succeeds. On the other, the cold logic of the machine that runs and does not get tired. Nor does research and advances in technology, which should be aimed at improving people's quality of life, always express the interests of citizens. However, some of the recent advances, due to their surprising development, arouse general interest and are sure to be of some relevance.

### 2.3. Beyond Digital

The challenge is to reaffirm a technology to build the digital era. Either priorities are set or a messy set of good wishes will lead to something that is not going to benefit either the scientific community or the citizens and the societies. Therefore, among the great objectives (robots that amaze us and machines that train themselves through the called "deep learning"), it is worth highlighting some priorities. Today, companies and research centers dispute the world hegemony in these techniques. Meanwhile, in our countries it would be desirable to generate the following paradigm: innovative activity is developed around specific problems. In terms of health, we can think of some proposals that should illuminate the way to go towards more preventive medicine. For example, in the early detection of diseases. Additionally, inclusive technological tools help people with disabilities and improve the quality of life of the elderly. In conclusion, hospital medicine has always been a favorable niche for colonization by machines, and physicians are voracious consumers of technology. It is true that artificial intelligence (AI) frees us from many repetitive tasks, but it is not going to replace professionals, at least not in the foreseeable future. In that sense, he is in diapers. As the philosopher would say: "In all cases the directing element must be the human person; it is about how to guide the human use of these systems to solve problems". Many of the innovations that appear frequently in the media, such as the robot surgeon and the mini robot that will clean the plaques of the arteries, are under investigation and have not passed the needed validation phases. AI will have to be developed much more before robots replace doctors [1].

### 3. Present

CAD and radiomics represent state-of-the-art clinical uses of digital imaging. They combine image processing, feature engineering, and statistical classification to produce image analysis tools. Their applications on lung and colorectal cancer, respectively, can illustrate the importance of these techniques in modern imaging.

### 3.1. CAD in Lung Cancer Diagnosis

When diagnosing lung cancer from nodules, firstly we need to detect the nodule in the image and then decide whether its radiological features are suspicious of malignancy or not. This process can determine whether a radiological follow-up or even an invasive

surgery is needed for the patient, thus having a huge impact on the outcome, and making it really important to obtain a good performance in the whole process. CAD systems try to replicate this process in which the radiologist takes part by detecting nodules, classifying them, or even both (Figure 2).

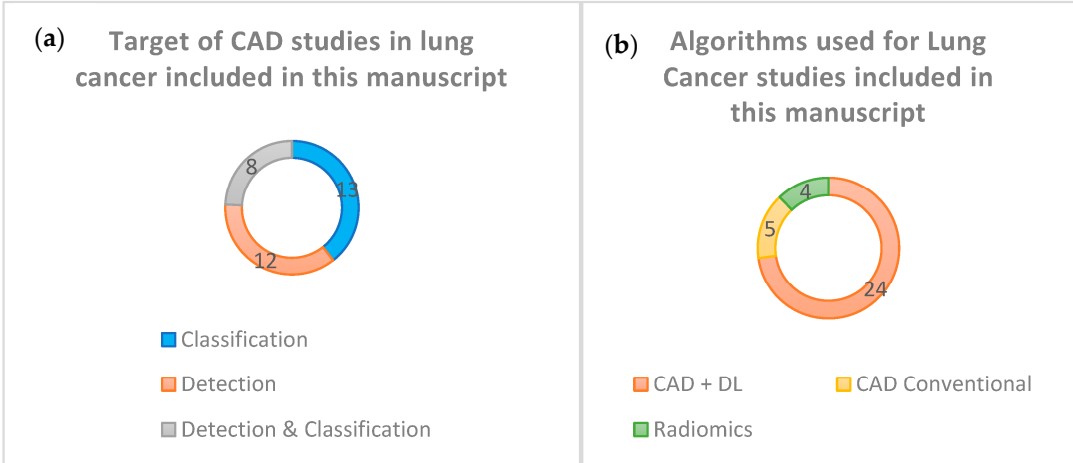

**Figure 2.** (**a**) Target of CAD studies about Lung Cancer included in this manuscript. (**b**) Algorithms used for Lung Cancer studies included in this manuscript.

### 3.1.1. Algorithms Proposed for Nodular Detection

We generally find two types of CAD systems for the detection of nodules in a CT image: conventional methods and Deep Learning based methods. Conventional methods rely heavily on hand-crafted features that are recognized in the image and thus can affect the generalizability of the classifying algorithms used (which usually classify the finding either as a nodule or a non-nodule finding). The conventional CAD models for nodular detection collected in Table 1 are three: Amer HM et al. [8], Gu Y. et al. [9], and Wagner A-K et al. [10]. Although they obtain good results, their performance could decrease sharply when using larger databases, since all three use a small number of scanners, and two of them make use of hospital databases that are not publicly available. The main reason why these models may experience a decrease in performance would presumably be the high variability of the nodules and the limitations of the human-established features, which are both disadvantages inherent to this type of model.

On the other hand, we have the DL-based models which are the rest. The most important difference between standard methods and these ones is that the latter, by supplying a large amount of medical imaging data, automatically learns the features that differentiate the relevant findings from what we would call normal. The nodule detection takes place in a process that usually includes establishing a database with correctly labeled results (according to the standard of reference) that the model is expected to obtain, choosing the learning type that best suits our model based on the input and output we are able to provide to the model, and dividing the database into subsets for training, validating and testing the model. Regarding the DL models that appear in the table, Huang X. et al. [11] propose a 2D convolutional neural network (CNN) model in contrast to most modern models, which use 3D CNN. Huang W. et al. [12] propose the use of Amalgamated-CNN. Li L et al. [13] use a DL-CNN model trained in a private database, using the results of the double reading of radiologists combined with the CAD model as the Standard of Reference, that is, a panel of expert radiologists that have the complementary information of the model itself. This DL algorithm demonstrated superior sensitivity, and therefore, detected more nodules than radiologists, as well as comparable performance in distinguishing solid nodules from ground glass opacity nodules, with the disadvantage of a high false positive rate. Tang H. et al. [14] use a U-Net-like 3D-Faster R CNN algorithm and a 3D classifier to reduce false positives. Zheng S et al. [15] make use of MIP images (Maximum Intensity

Projections) in their model and conclude that this achieves notable benefits when using CNN to detect lung lesions, especially the smallest ones. Gong L et al. [16] affirm that the success of their model comes from three key points: that the model is based on 3D-DCNN, the use of a 3D-region proposal network (RPN) with a U-Net type structure and subsequent reduction in FP, and the use of Residual Networks Squeeze and Excitation (SE) type to accelerate the training process and the accuracy of nodular detection. Tan J. et al. [17] make use of a DL-CNN model that takes into account features such as shape and texture and propose to study the implementation of rather traditional CAD features such as curvature or entropy in future works, as well as clinical ones (e.g., age, sex, race, tobacco, etc.). Finally, the model by Tran GS. et al. [18] represents an exception compared to the rest of the studies included in the table since they propose a 2D-DCNN model (named LdcNet) that classifies the candidate findings into nodule and non-nodule but requires the prior detection of these findings which they obtain from the LIDC database. Therefore, the model detects nodules conditioned to have candidate findings but at the same time it cannot differentiate malignant from benign, which is a reason why it is included in this section. In their model, they use Focal Loss in order to improve the accuracy of the classification of the candidate findings.

### 3.1.2. Algorithms Proposed for Nodular Classification

The classification process although similar to the one in detection presents two major differences: the first one is that information about nodule location is supplied to the algorithm (usually databases used for this purpose provide coordinates about interesting findings in the image), and the second is that the model does not differentiate between nodule and non-nodule findings, but rather differentiates between malignant and non-malignant nodules.

Of all the models found in Table 1, only Tammemagi M et al. [19] propose the use of conventional CAD for classification purposes, using two models: one volume based and one diameter based with similar results, although slightly higher in the volume one. The rest of the CAD models make use of DL, such as Xie Y. et al. [20], who propose a semi-supervised model considering that the use of unlabeled data is not only effective for the training of the DL models but also critical to carry out a hypothetical lung nodule screening system. Zhao X. et al. [21] compare 11 different algorithms based on DL-CNN and propose one based on Transfer Learning after showing that it showed promising results. Da Silva et al. [22] study three different CNN architectures of which the third one obtains the best results and encourages finding new applications of Deep Learning in computer-assisted diagnosis. Kailasam SP et al. [23] tested 36 combinations of which the algorithm that combines the Support Vector Machine (SVM) classifier with an Extended Histogram of Oriented Gradients (ExHOG) and CNN achieves the highest accuracy when classifying. Wu P et al. [24] test different methods from which a 50-layer Residual Network based model obtains the best performance, and its main defect is that it requires a long period of training when it has to deal with a large number of pulmonary CT images, something they will try to solve in future works. Zhang S. et al. [25] propose two models, a Multi-Channel Multi-Slice 2D CNN and a Voxel- Level 1D CNN, the latter is especially useful in small and poorly balanced data sets. At the same time, they question the validity of these openly used databases used to discern between malignant and benign nodules, such as that of LIDC-IDRI, as the Standard of Reference in the latter is not a pathology study but a consensus among four radiologists in which global consensus is not necessary, and therefore, only 25% were classified as nodules by all four while the rest were not. As a solution to improve this type of bias that may affect the design of the classification models, it is proposed to use data sets whose reference standard is pathology study.

**Table 1.** CAD on lung cancer diagnoses.

| CAD Model | Method | Function | Data Base | N° Scanners | N° Nodules | Sensitivity | Specificity | FPs/Scan | Accuracy | AUC | Year |
|---|---|---|---|---|---|---|---|---|---|---|---|
| Amer HM et al. [8] | CAD conventional | Detection | ELCAP | 40 | NA | 100% | 99.20% | NA | 99.60% | NA | 2019 |
| Gu Y. et al. [9] | CAD | Detection | LIDC-IDRI | 154 | 204 | 87.81% | NA | 1.057 | NA | NA | 2019 |
| Wagner A-K et al. [10] | CAD conventional | Detection | Private | 100 | 106 | 87.00% | NA | NA | NA | NA | 2019 |
| Huang X. et al. [11] | CAD + DL + CNN | Detection | LUNA16 | 888 | NA | NA | NA | 1 4 | 94.60% | NA | 2019 |
| Huang W. et al. [12] | CAD + DL + CNN | Detection | LUNA16; Ali Tianch | NA | 1795 | 81.70% 85.10% 88.30% 90.70% | NA | 0.125 0.25 1 4 | 91.40% | NA | 2019 |
| Li L et al. [13] | CAD + DL + CNN | Detection | Private | 346 | 812 | 86.20% | NA | 1.53 | NA | NA | 2019 |
| Tan H. et al. [14] | CAD + DL + CNN | Detection | LUNA16 | 888 | 1186 | 86.4% 85.2% | NA | 4 1 | NA | | 2019 |
| Zheng S et al. [15] | CAD + DL + CNN | Detection | LUNA16 | 888 | 1186 | 92.70% 94.20% | NA | 1 2 | NA | NA | 2020 |
| Gong L et al. [16] | CAD + DL + CNN | Detection | LUNA16 | 888 | 1186 | 93.60% 95.70% | NA | 1 4 | NA | NA | 2019 |
| Tan J et al. [17] | CAD + DL + CNN | Detection | LIDC-IDRI | 208 | NA | 80.10% 94.0% | NA | 1.89 4.01 | NA | NA | 2019 |
| Tran G. et al. [18] | CAD + DL + CNN | Classification | LUNA16 | 888 | 1186 | 96.00% | 97.30% | NA | 97.20% | 0.9820 | 2019 |
| Tammemagi M et al [19] | CAD conventional (volume) | Classification | NSLT | 3680 | 6009 | 75.00% | 75.00% | NA | NA | 0.8210 | 2019 |
| Tammemagi M et al [19] | CAD conventional (diameter) | Classification | NSLT | 3680 | 6009 | 75.00% | 75.00% | NA | NA | 0.810 | 2019 |
| Xie Y. et al. [20] | CAD + DL + CNN | Classification | LIDC-IDRI | 1018 | 1945 | 84.94% | 96.59% | NA | 92.53% | 0.9581 | 2019 |
| Zhao X. et al. [21] | CAD + DL + CNN | Classification | LIDC-IDRI | 1018 | 368 | 91.00% | NA | NA | 88.00% | 0.94 | 2019 |
| Da Silva et al. [22] | CAD + DL + CNN | Classification | LIDC-IDRI | 833 | 1296 | 79.40% | 83.80% | NA | 83.30% | NA | 2020 |
| Kailasam SP et al [23] | CAD + DL + CNN | Classification | LIDC-IDRI | NA | 467 | NA | NA | NA | 95.32% | NA | 2019 |
| Wu P et al. [24] | CAD + DL + CNN | Classification | LIDC-IDRI | 1018 | NA | 97.70% | 98.35% | NA | 98.23% | NA | 2020 |
| Zhang S. et al. [25] | CAD + DL + CNN | Classification | LIDC-IDRI | 1018 | NA | NA | NA | NA | 97.04% | NA | 2019 |
| Liu A et al. [26] | Radiomics-CT | Classification | Private | 263 | 263 | NA | NA | NA | NA | 0.809 | 2020 |
| Mao L et al. [27] | Radiomics-LDCT | Classification | Private | 98 | 98 | NA | NA | NA | 89.80% | 0.97 | 2019 |

**Table 1.** *Cont.*

| CAD Model | Method | Function | Data Base | N° Scanners | N° Nodules | Sensitivity | Specificity | FPs/Scan | Accuracy | AUC | Year |
|---|---|---|---|---|---|---|---|---|---|---|---|
| Xu Y [28] | Radiomics-CT | Classification | Private | 373 | 373 | 89.00% | 74.00% | NA | 77.00% | 0.84 | 2019 |
| Zhou Z et al. [29] | Radiomics-LDCT | Classification | LIDC-IDRI | 1018 | 1226 | 85.80% | 90.70% | NA | 88.90% | 0.935 | 2019 |
| Asuntha A et al. [30] | CAD + DL + CNN | Detection and Classification | LIDC-IDRI | 1018 | NA | 97.93% | 96.32% | NA | 95.62% | NA | 2020 |
| Bhandary A et al. [31] | CAD + DL + CNN | Detection and Classification | LIDC-IDRI | 1018 | NA | 98.09% | 95.63% | NA | 97.27% | 0.996 | 2020 |
| Bansal G et al. [32] | CAD + DL + ResNet | Detection and Classification | LUNA16 | 888 | NA | 87.10% | 89.66% | NA | 88.33% | 0.88 | 2020 |
| El-Bana S et al. [33] | CAD + DL + TL | Detection and Classification | LUNA16; KAGGLE | 888; 1397 | NA | 96.40% | 99.40% | 0.6 | 97.00% | NA | 2020 |
| Masood A et al. [34] | CAD + DL + CNN | Detection | LUNA16 | 888 | NA | 81.20% 97.80% 98.53% 98.66% | NA | 0.125 1 4 8 | NA | NA | 2019 |
| Masood A et al. [34] | CAD + DL + CNN | Detection | LIDC-IDRI; ANODE09 | 1190 | NA | 98.40% | NA | 2.1 | NA | NA | 2019 |
| Nasrullah N et al [35] | CAD + DL-CNN | Detection and Classification | LIDC-IDRI | 1200 | 3250 | 94.00% | 90.00% | NA | 91.13% | 0.99 | 2019 |
| Nasrullah N et al [35] | CAD + DL-CNN + Biomarkers | Detection and Classification | LIDC-IDRI | 1018 | 2562 | 93.97% | 89.93% | NA | 88.79% | NA | 2019 |
| Shanid M et al. [36] | CAD + DL + DBN | Detection and Classification | LIDC-IDRI | 1018 | NA | NA | NA | NA | 96.00% | NA | 2019 |
| Zhang C. et al. [37] | CAD + DL + CNN | Detection and Classification | LUNA16; KAGGLE | 757 | 855 | 84.4% | 83.00% | NA | 83.70% | 0.803 | 2019 |

Some authors defend that the radiological information obtained from CT may be the key to the correct classification of nodules as benign or malignant. They argue that the aforementioned classification models are based on the statistical study of various image values without taking into account relevant clinical data, and without a detailed analysis that could demonstrate correlations between phenotype and histology or genetics that would provide valuable information about these pulmonary nodules. Liu A et al. [26] investigate and develop a radiological nomogram to assist in the diagnosis of potentially malignant pulmonary nodules. This would constitute a non-invasive diagnostic tool that helps make clinical and surgical decisions. They bet on the use of a huge number of radiological features as a determining factor in the best performance of the model. In this study, the private database of a hospital is used, establishing as an inclusion criterion that patients have or are going to have a definitive histological diagnosis. This brings reliability to the data set studied as well as new biases, such as the possible fact that presumably healthy patients do not choose to undergo a biopsy or surgery. Mao L. et al. [27] affirm that despite the amount of available knowledge nowadays, there are almost no studies that specifically focus on the usefulness of radiomics in predicting malignancy of small and solid nodules in screening for lung cancer with LDCT. Small nodules have a much lower number of specific features compared to large ones, and malignant solid nodules progress much faster than subsolid nodules. For these reasons, the purpose of the study is to develop a predictive radiological model to diagnose small solid lung nodules (6–15 mm), since it is something that has not been exhaustively investigated. Xu Y. et al. [28] used the scanners of 373 patients who would later undergo surgery or CT-guided percutaneous biopsy, thus obtaining a pathological confirmation of the nodule. They propose a novel non-invasive predictive method for diagnosing lung nodules based on the eighth edition of the TNM system. After studying some 1160 possible radiological features to use in the model, they concluded that depending on the size of the pulmonary nodule, some have more weight than others in the classification as malignant or benign. Thus, in nodules of 2 cm or less, the textures are the main contributors to the correct classification, as are the shapes between 2 and 3 cm. Zhou Z et al. [29] propose a Multi-Objective based Feature Selection (MO-FS) algorithm to select the most important radiographic features for the classification, demonstrating that the set developed by this algorithm achieves superior performance compared to the rest of the algorithms commonly used with this function.

### 3.1.3. Algorithms Proposed for Detection and Nodular Classification

Asuntha A et al. [30] propose a novel Fuzzy Particle Swarm Optimization (FPSO) algorithm used to choose the most specific nodular feature after extracting data from the image so that the computational complexity of CNN is reduced. Bhandary A et al. [31] use the LIDC-IDRI database to propose two models: one for CT and one for chest radiography. With the CT model, they achieve a sensitivity of 98.09%, a specificity of 95.63% and an accuracy of 97.27%. Bansal G et al. [32] propose a DL model using ResNet, for the detection and classification of the pulmonary nodule. They consider the process of differentiating a malignant nodule from a benign one difficult work due to the minuscule differences that may exist between the two, and for this, they propose 18 morphological features to take into account when classifying the nodule such as perimeter, orientation, area, eccentricity, etc. El-Bana S et al. [33] use a Deep learning model combined with a Transfer Learning method in order to save time when training their new model, taking another already trained and whose obtained values are known and re-applying it to the new categories concerned in the study. Masood A et al. [34] propose a 3D-DCNN model to detect and classify nodules greater than or equal to 3 mm that achieves a higher sensitivity and FP/scanner than the other models with which it is compared. This model classifies findings into nodules (benign or malignant) and non-nodules. In micronodules of less than 3 mm, the performance is lower, which is why they conclude that in future works it will be an objective to achieve their detection while maintaining a good relationship between sensitivity and FP by a scanner. Nasrullah N et al. [35] propose a three steps 3D Faster

Region-CNN system in which they first detect lung nodules, second classify them as benign or malignant, and thirdly the results obtained are combined with multiple clinical factors such as family history, age, tobacco, biomarkers, size and location of the nodule. Shanid M et al. [36] develop a CAD model of DL using the Deep Belief Network (DBN), another type of Deep Neural Network. Zhang C. et al. [37] propose a DL algorithm that performs the detection and classification processes in a unified way in contrast to most studies, which apply a sequential algorithm since this reduces the possibility of accumulating errors. From the databases of various centers, the model was validated and a subset of 50 patients was obtained where the model was tested, comparing it with 25 physicians (including radiologists, thoracic surgeons, and pulmonologists) obtaining better detection performance and classification of pulmonary nodules.

### 3.2. Radiomics and Personalized Medicine

Radiomics is a discipline of medicine that is based on the massive extraction of quantitative data from medical images [38]. The fundamental idea is that images are more than pictures and that there is an enormous amount of information that is not visible to the eyes of radiologists which can reveal underlying pathophysiological processes. These quantitative data obtained from medical images will vary depending on whether there is an alteration, thus being able to obtain a "radiomic phenotype" of a certain underlying alteration.

The term radiomics was coined in 2012, and in its early days, its development focused on the field of oncology [38]. It is possible to use it in any disease whose study uses imaging, and in many different imaging modalities such as computed tomography (CT), magnetic resonance imaging (MRI) or positron emission tomography (PET). It was designed to be another tool to be used in daily clinical practice at different levels of the healthcare system. Thanks to the information obtained through radiomics, together with the relevant clinical data of a patient, it would be possible to assist in the detection of tumors, the diagnosis of diseases, as well as to provide information on prognosis, response to treatments, or evaluation of the state of a disease.

The ending "-omics" derives from the field of molecular biology, where it is used for the detailed study of biomolecules (genomics, proteomics, etc.). This termination applies to other fields where a large amount of data is handled, as in the case of the quantitative data we obtain from radiological images (radiomics) [38]. The analysis of these data is what allows us to generate hypotheses. In this way, radiological information offers new advantages in the study of patients' diseases but also presents numerous limitations such as the handling of such large amounts of data.

As mentioned above, this technology may have a key role in the treatment of tumor pathology. One of the great problems in treating cancer is the great heterogeneity that malignant lesions have at the genetic, physiological, and phenotypic levels. This heterogeneity can develop both intratumorally and between metastases of the same cancer, and can even develop over time, so an initially effective treatment can cease to be so in a matter of months due to these alterations in pharmacological targets. In the current context of medicine, with its evolution increasingly towards totally personalized medicine, knowing the mechanisms of tumor resistance to therapy is mandatory to choose a focused and effective treatment [38]. This task requires an early detection method that is reproducible and minimally invasive, in addition to the development of new drugs, to counteract these drawbacks. Radiomics could offer a solution to this challenge, allowing the detection of relevant changes at the genomic level in the tumor (radiogenomics), provided that these are expressed as changes at the radiological level [39].

Radiomics definition workflow is independent of the disease under study and consists of five consecutive steps (Figure 3):

**Figure 3.** Radiomics workflow.

### 3.2.1. Obtaining Images

The first step is to obtain a biomedical image that must be integrated into a database from which to extract hypotheses. However, in routine clinical image acquisition, there is a wide variation in imaging parameters and reconstruction algorithms. These imaging issues can create difficulty in comparing results obtained across institutions with different scanners and patient populations. In order to avoid this, all image acquisition and reconstruction must be standardized [38,39].

### 3.2.2. Pre-Processing

Pre-processing tools can improve image quality by reducing the noise of the image or emphasizing different characteristics at different scales. However, their application must be careful because these tools can also alter the radiomics signature. It is also important to consider that the volumes under study can be single or multiple (metastases). These volumes are usually heterogeneous and can be studied by dividing them into habitats, which represent phenotypic differences within the same tumor.

### 3.2.3. Segmentation

In this phase, the parts of the image that were considered of interest (called "regions of interest" or ROIs) are separated from the rest of the image and the data of these relevant segments are obtained [40,41]. The delineation of this ROI can be conducted manually, semi-automatically, or automatically, in 2D or 3D. This is a critical phase because many tumors can have edges indistinguishable from normal tissues, and the data obtained in this phase can experience variations [40]. Manual segmentation is more laborious and presents greater interobserver variability, on the other hand, in tumor pathology it is often almost the only alternative due to how poorly delimited the lesion usually is. Manual segmentation is usually performed by the radiologist in a slice-by-slice manner, and the ideal is to include the highest percentage of the lesion in the study [40,41]. It is considered the gold standard, but it requires a considerable investment of time. Currently, there are very refined semi-automatic 3D-segmentation tools (Figure 4), which allow segmentation through artificial intelligence guided by the radiologist himself. In this way, in a few seconds, the entire tumor volume in the segmented region can be included in a supervised way. Automatic segmentation in oncology has a limited application because the heterogeneity and the poorly differentiated borders of many tumors mean that the segmentation usually includes areas in the de ROI where there is no tumor, with the consequences that this would entail in the extraction of features [41].

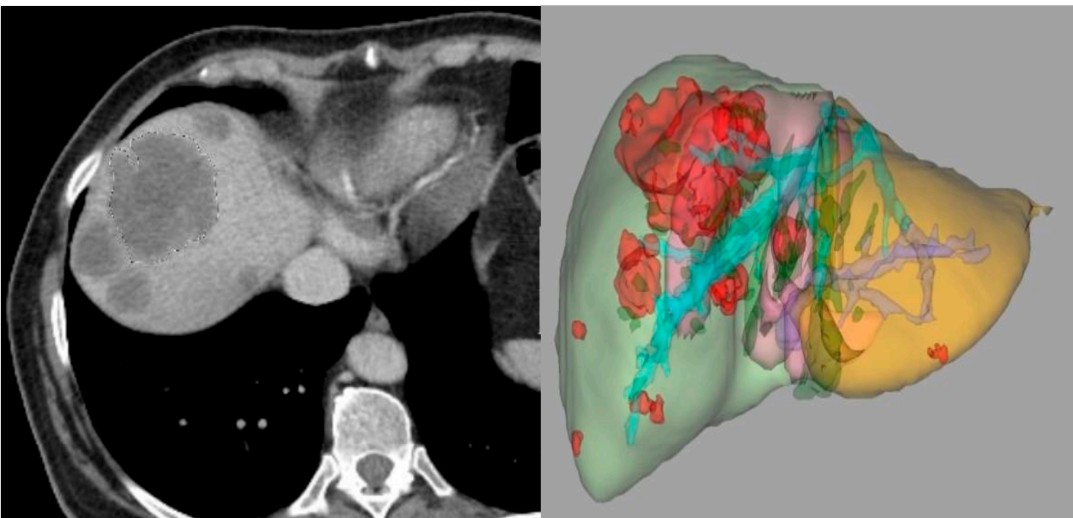

**Figure 4.** Three-dimensional semi-automatic segmentation of liver metastases of colorectal cancer.

### 3.2.4. Feature Extraction and Classification

There is a multitude of characteristics from which quantitative data can be extracted (morphological characteristics, intensity, texture, etc.). The data obtained from the extraction of characteristics can be processed later to increase the capacity of discrimination, establish the most identifying feature and reduce possible interference (for example, redundancy). The effort made in search of the standardization of feature extraction has led to the creation of initiatives such as the Imaging Biomarker Standardization Initiative (IBSI), which standardized the extraction of a set of 169 radiomics features [42]. The data obtained from the images can be added to clinical or genomic data from the patient which would be useful, so the integration of this information would offer the possibility of obtaining more specific conclusions and of greater value, provided that all the data added to the database are reliable.

### 3.2.5. Analysis of Data

The power of radiological information to offer hypotheses and possible correlations lies in the amount of data available. The more data, the greater the analysis that can be performed and its power to infer correct correlations (for example, to determine if a tumor has a specific mutation or not). Data analysis consists of two phases: the first phase uses a series of data to create a classification and/or regression model with them. The second phase uses the data of the patients under study, in which it will use the model created to make predictions about them. In the creation of the classification model, in addition to the input data, typically the output data that the model aims to predict (benign or malignant, presence or absence of a mutation, etc.) must also be entered [41]. Ideally, the data to create the model should be from another institution or at least different from what will later be used to apply the created model.

### 3.3. Radiomics and Radiogenomics in Colorectal Cancer

Currently, in the process of diagnosing and treating colorectal cancer (CRC), genomic analysis has a fundamental role. Radiogenomics uses radiomics data to develop a predictive model of tumor genomics [41]. In the future, radiogenomics may represent a complementary model to biopsy, the so-called "virtual biopsy". Genomic diagnosis by biopsy may lose key tumor features given the intratumoral heterogeneity. In this scenario, radiogenomics may define and locate tumor habitats susceptible to being biopsied because they present a greater probability of mutation of certain genes. In patients with CRC, the presence of mutations in the RAS-RAF-MEK-ERK signaling pathway causes these tumors to present resistance to anti-EGFR monoclonal treatments (cetuximab and

panitumumab) [38]. However, the articles about radiomics in CRC do not only focus on the genetic determination. As can be seen in the literature, there are numerous studies about the therapy response, prognostic predictions, survival determination, microsatellite status, prediction of metastatic disease, etc. The majority of radiomics studies in CRC used a CT-based radiomics model because CT is the imaging technique of choice for staging in CRC patients (Figure 5).

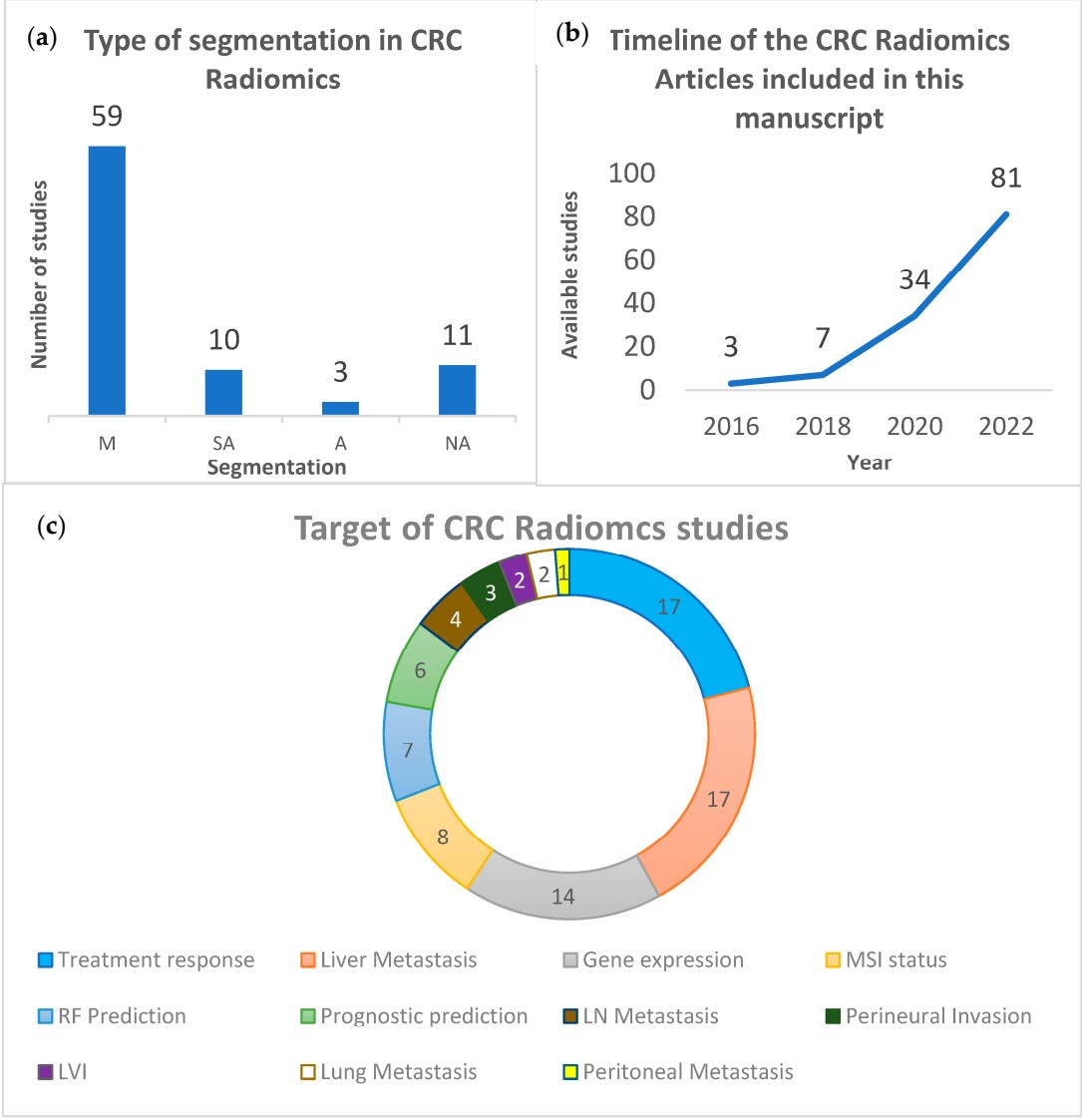

**Figure 5.** (**a**) Number of CRC-Radiomics articles that used manual (M), semi-automatic (SA) and automatic (A) segmentation. (NA: Not Available). (**b**) Timeline of the CRC-Radiomics articles published since 2016 included in this manuscript. (**c**) Number of CRC-Radiomics articles according to their target.

CT-Based Radiomics/Radiogenomics in Colorectal Cancer

Regarding radiomics and radiogenomics models that use CT images (Table 2), most articles are focused on the prediction of treatment response and the development of liver metastasis. Manual segmentation (M) is usually the type of segmentation used in the radiomics assessment of CRC with only a few papers based on a semi-automatic segmentation (SA). The main studies in the literature on this topic are retrospective (R), reflecting a need for prospective studies (P). There is also a wide variation in both the number of patients (N) and features extracted (RF) included in the studies. Many of the studies only

reflect the features that they considered significant. Other studies reflect all the features extracted and the posterior features which were considered. In the results and conclusion, radiomics features show good results, but it can be seen in many studies that the results improve when the radiomics information and other clinical characteristics (such as CEA, T staging, etc.) are incorporated.

Concerning the prognostic value of radiomics features in CRC, in the survival prediction, Xue et al. concluded that a combined nomogram with a radiomics signature based on CT images and clinical predictors improves the predicted accuracy of the overall survival (OS) in CRC patients [43]. Similar results were reported by Huang et al. which achieved a hazard ratio (HR) of 6.670, 2.866 and 3.342 in the training and two external validation cohorts in the association between radiomic features and OS [44]. Dercle et al. obtained an HR between 3.93 and 21.04 in the prediction of OS using three different radiomics features [45]. In a different study, Badic et al. compared the correlation between texture features obtained from contrast-enhanced (CE) and noncontrast enhanced (NCE) CT images and OS and concluded that some of the second and third-order features were associated with patient's survival [46]. Finally, Mühlberg et al. evaluated the prediction of 1-year survival in patients with metastatic CRC with a comparative analysis of quantitative imaging biomarkers based on the geometric analysis (GMS) of the whole liver tumor burden (WLTB), in comparison to predictions based on the tumor burden score (TBS), WLTB alone, and a clinical model [47].

The combination of radiomics features and clinical information for prognostic prediction may represent a useful tool in the assessment of CRC patients. Li et al. found that this combination can predict the presence of distance metastasis and 3-year OS in these patients [48]. Zhao et al. also developed a combined model (radiomics-clinical) for 3-year OS in CRC patients treated with targeted therapy [49]. Ye et al. evaluated radiomics features combined with clinical characteristics in patients with CRC liver metastases before and after chemotherapy (called Delta radiomics). They reported an AUC of 0.871 for the training cohort and 0.745 for the validation cohort in the prediction of 1-year progression-free survival (PFS) [50].

Radiomics have also been used to predict tumor therapy response. Rabe et al. found eight radiomic features using a 3D semi-automatic segmentation model which were associated with chemotherapy response in non-necrotic liver metastasis of CRC [51]. Cai et al. also concluded that radiomics scores based on five radiomics features selected using the LASSO algorithm are an independent prognostic factor of tumor response [52]. Different studies have found similar results. Defeudis et al. had promising results for the prediction of the response of liver metastases from CRC to first-line chemotherapy [53]. Lutsyk et al. studied the association between radiomics features and complete response after neoadjuvant chemoradiation [54]. Bibault et al. using deep learning-based radiomics, had an accuracy of 0.80 in predicting complete neoadjuvant chemotherapy response [55]. Zhang et al. were able to predict tumor resistance to therapy with a radiomics model [56]. In a different study using Delta-radiomics, Giannini et al. obtained sensitivity, specificity, and positive and negative predictive values (PPV and NPV) between 0.85 and 0.99 in the prediction of non-response first-line chemotherapy in CRC patients with liver metastases (LmCRC) [57]. Vandendorpe et al. obtained an ACU of 0.70 for the prediction of downstaging after chemotherapy [58]. Zhuang et al. obtained an AUC of 0.997 using a combined model with clinicopathological information and radiomics features [59]. Wang et al. with a combined model for the prediction of locoregional failure free survival (FS), obtained an AUC of 0.68, and 0.64 OS [60]. Dercle et al. obtained an AUC of 0.80 and 0.72 for predicting the response to anti-EGFR [61]. Yuan et al. obtained an accuracy of 0.839 in the prediction of pathological complete response (TRG 0) vs. moderate, partial and poor response (TRG 1, 2 and 3, respectively) [62]. Bonomo et al. obtained an AUC of 0.65 in the prediction of good response after neoadjuvant chemotherapy [63]. Finally, Fan et al. obtained significant results in predicting postoperative recurrence risk [64], and Badic et al. also studied the prediction of recurrence after surgery. The best balanced accuracy (BAcc) was 0.78 [65].

**Table 2.** CT-based Radiomics/Radiogenomics on CCR.

| Author | Year | Type | N | Target | ROI | RF | Results | Conclusions |
|---|---|---|---|---|---|---|---|---|
| Xue [43] | 2022 | R | 121 | Prognostic prediction | NA | NA | C-Index 0.782, 0.721 and 0.677 | Combined nomogram (radiomic-clinical) improves the accuracy of survival prognostic. |
| Huang [44] | 2022 | R | 512 | Prognostic prediction | M | 45 | HR 6.670, 2.866 and 3.342 | Radiomic features could be used for predicting OS |
| Dercle [45] | 2022 | R | 1584 | Prognostic prediction | NA | NA | HR incremented from 3.93 to 21.04 using RF | Combined model with radiomic features can provide information and improve decisions |
| Badic [46] | 2019 | R | 61 | Prognostic prediction | SA | 21 | rs max = 0.49 for first order features rs max = 0.770 for some second and third order features | Some radiomics features with moderate correlations between NCE-CT and CE-CT images |
| Mühlberg [47] | 2021 | R | 103 | Prognostic prediction | A | >1500 | AUC 0.73 and 0.76 for 1-year survival prediction | Geometric distribution and RF yield prognostic information |
| Li [48] | 2020 | R | 148 | Prognostic prediction | M | 17 | AUC of 0.842 and 0.802 for the combined model The combined model showed better prediction of OS | Combined model can help to predict distant metastasis |
| Zhao [49] | 2021 | R | 80 | Treatment response | M | 48 | C-index of 0.8335 and 0.9182 | RF are prognostic factors and predictive markers of OS |
| Ye [50] | 2022 | R | 139 | Treatment response | M | 1316 | AUC 0.871 and 0.745 for PFS | Combined model had better prediction results |
| Rabe [51] | 2022 | R | 29 | Treatment response | SA | 175 | AUC 0.80; S 0.73; Spec 0.79 | 8 RF had a significant association with treatment response |
| Cai [52] | 2020 | R | 381 | Treatment response | M | 85 | AUC of 0.74 and 0.82 | Radiomics score is an independent prognostic factor |
| Defeudis [53] | 2021 | R | 92 | Treatment response | M | 75 | S 0.61; Spec 0.60; PPV 0.57; NPV 0.64 | Promising results for determining the chemotherapy response |
| Lutsyk [54] | 2021 | R | 140 | Treatment response | M | 850 | Acc 0.63 405 RF were different ($p < 0.001$) between groups | Imagine features can help to determine complete and non-complete response |
| Bibault [55] | 2018 | R | 95 | Treatment response | M | 1683 | Acc of 0.80 | DL with clinical and RF can predict complete neoadjuvant chemotherapy response |
| Zhang [56] | 2022 | R | 215 | Treatment response | M | 275 | AUC of 0.92 and 0.89 | CT-based radiomics could be helpful in the treatment planning |
| Giannini [57] | 2022 | R | 301 | Treatment response | M | 107 | S 99–94%, Spec 95–99%, PPV 85–92%, NPV 90–87% | Delta radiomics signature was able to predict non-response |
| Vandendorpe [58] | 2019 | R | 121 | Treatment response | M | 36 | AUC of 0.70 predicting downstaging OR 13.25 for Radscore as independent factor | This prognostic score may lead to improve the treatment |
| Zhuang [59] | 2021 | R | 177 | Treatment response | M | 1218 | AUC 0.997 and 0.822 for prediction of CR | CT-based radiomics can help in the prediction of complete chemotherapy response |

**Table 2.** *Cont.*

| Author | Year | Type | N | Target | ROI | RF | Results | Conclusions |
|---|---|---|---|---|---|---|---|---|
| Wang [60] | 2022 | R | 191 | Treatment response | M | 1130 | AUC of 0.68 for locoregional failure FS. AUC of 0.64 for OS | CT-based radiomics can predict the NAR punctuation and the survival outcomes |
| Dercle [61] | 2020 | R | 667 | Treatment response | M | 3499 | AUC 0.80 and 0.72 for sensitivity to anti-EGFR AUC of 0.59 and 0.55 for chemotherapy response | RF can help in the early prediction of the success of treatment with Cetuximab |
| Yuan [62] | 2020 | R | 91 | Treatment response | NA | 8 | Acc of 83.9% differentiating TRG 0 vs. TRG 1–3 | Promising results for predicting pathologic complete response. |
| Bonomo [63] | 2022 | R | 201 | Treatment response | M | 1150 | AUC of 0.65 on prediction of GR | CT-base radiomics has potential predictive ability for identifying patients with GR |
| Fan [64] | 2021 | R | 299 | Treatment response | SA | 1561 | OR de 239,993 ($p < 0.001$) for recurrence risk AUC of 0.954 and 0.906 | Radiomic signature is an independent risk predictor and a non-invasive biomarker |
| Badic [65] | 2022 | R | 193 | Treatment response | SA | 88 | BAcc was 0.78 for recurrence prediction | CT-based radiomics had a good predictive performance of recurrence |
| Hong [66] | 2022 | R | 292 | Risk factors prediction | NA | NA | AUC 0.799 for combined model AUC of 0.679 for CT staging only | Combined model can improve the detection of high-risk colon cancer |
| Ge [67] | 2020 | R | 225 | Risk factors prediction | M | 396 | AUC 0.93 for the differentiation between mucinous and non-mucinous CRC | CT RF could be utilized as a noninvasive biomarker to identify MA from NMA patients |
| Hu [68] | 2016 | *p* | 40 | Risk factors prediction | M | 775 | 496 RF showed high reproducibility 225 shoed median reproducibility 54 showed low reproducibility | Some RF showed stability and could be used for treatment monitoring |
| Dou [69] | 2022 | R | 32 | Risk factors prediction | M | 125 | 3 parameters are associated with high and low risk group of metastases | Some RF could be used to help the T staging |
| Liu [70] | 2021 | R | 134 | Risk factors prediction | M | 854 (16) | AUC 0.945 and 0.754 for radiomic signature AUC 0.981 and 0.822 with multiscale nomogram | The multiscale nomogram could be used to facilitate the individualized preoperatively assessing metastasis in CRC patients |
| Huang [71] | 2018 | R | 366 | Risk factors prediction | M | 10959 | AUC of 0.8122 and 0.735 in discrimination between high and low CRC grade. | This radiomics signature can help with personal treatment |
| Liang [72] | 2016 | R | 494 | Risk factors prediction | M | 16 | AUC 0.792 and 0.708 | Radiomics signature can discriminate between stages I-II and III-IV |
| Badic [73] | 2019 | R | 64 | Gene expression | SA | 27 | ABCC2, CD166, CDKNV1 and INHBB genes has significant correlation with RF | Combined RF with genetic and pathological information can help to patient management |
| Chu [74] | 2020 | R | 163 141 | Prognostic prediction Gene expression | M | 12 | AUC 0.641 for prognostic prediction AUC 0.829 and 0.727 for CXCL8 | Combined model had better results. There are associations between RF and CXCL8 |

**Table 2.** *Cont.*

| Author | Year | Type | N | Target | ROI | RF | Results | Conclusions |
|--------|------|------|---|--------|-----|-----|---------|-------------|
| Huang [75] | 2022 | R | 71 | Prognostic prediction Gene expression | M | 1037 | 10 RF with AUC 0.46–0.56 for recurrence prediction. | Association RF-recurrence prediction. Association with some gene expression. |
| Hoshino [76] | 2022 | R | 24 | Gene expression | M | 1037 | AUC of 0.732 and 0.812 for predicting TBM status. S of 0.857, Spec of 0.600 and Acc of 0.682 | The accurate inference of the TBM status is possible using radiogenomics |
| Yang [77] | 2018 | R | 117 | Gene expression | M | 346 | AUC 0.869–0.829; S 0.757–0.686; Spec 0.833–0.857 | Radiomic signature based on CT is associated with KRAS/NRAS/BRAF mutations |
| Shi [78] | 2020 | R | 159 | Gene expression | SA | 851 | AUC of 0.95 and 0.79 for the combined model for distinguishing between wild type and mutant | Radiomics together with semantic features can improve non-invasive assessment of KRAS status of LmCRC |
| González-Castro [79] | 2020 | R | 47 | Gene expression | M | 30 | Acc of 0.83; Kappa index of 0.647; S of 0.889 and Spec of 0.75 for the prediction of KRAS mutation | RF based on CT images can predict the KRAS mutation status |
| Wu [80] | 2020 | R | 279 | Gene expression | M | 50 | C index of 0.719 for Radiomics; 0.754 for DL-radiomics; 0.815 and 0.932 for combined model (1st and 2nd cohorts) in the prediction of KRAS mutation | This is a model that incorporates standard radiomics with deep learning-based radiomics. |
| He [81] | 2020 | R | 157 | Gene expression | M | 1025 | AUC of 0.818 | CT-based radiomics can predict KRAS mutation. |
| Hu [82] | 2022 | R | 231 | Gene expression | M | 1316 | AUC was 0.8826 for arterial and venous phase model | CT-based radiomics has potential to predict KRAS mutation |
| Jang [83] | 2021 | R | 110 | Gene expression | NA | 378 | AUC of 0.73 radiogenomics model AUC of 0.63 DL model | Radiomics model obtained better results than deep learning |
| Xue [84] | 2022 | R | 172 | Gene expression | NA | 1018 | AUC of 0.75 and 0.84 (2D and 3D radiomics models) for the 8 selected RF; AUC of 0.92 for the combined nomogram | CT-Radiomics can predict KRAS mutations. Combined nomogram improves the results |
| Xue [85] | 2022 | R | 140 | Gene expression | NA | NA | AUC of 0.93 and 0.87 for the 5 best RF; AUC of 0.95 and 0.88 for a combined nomogram | CT-based radiomics is associated with BRAF mutation |
| Negreros-Osuna [86] | 2020 | R | 145 | Gene expression | M | 24 | Some RF were significantly different between BRAF mutant and wild-type ($p < 0.05$) Some RF were associated with better 5-year OS (HR 0.40) | RF can serve as potential biomarkers for determining BRAF mutation status and as predictors of 5-year OS |
| Fan [87] | 2019 | R | 119 | MSI status | SA | 398 | Radiomics: AUC 0.688; Acc 0.713; S 0.517; Spec 0.858. Clinical: AUC 0.598; Acc 0.632; S 0.371; Spec 0.825. Combined model: AUC 0.752; Acc 0.765; S 0.663; Spec 0.842 | CT-based radiomics are associated with MSI status |
| Li [88] | 2021 | R | 368 | MSI status | M | 1628 | AUC 0.79 and 0.73 | Combined model can predict MSI status |
| Ying [89] | 2022 | R | 276 | MSI status | M | 1037 | AUC 0.87 and 0.90 | Combined nomogram can predict MSI status |

**Table 2.** *Cont.*

| Author | Year | Type | N | Target | ROI | RF | Results | Conclusions |
|---|---|---|---|---|---|---|---|---|
| Chen [90] | 2022 | R | 837 | MSI status | NA | 10 | AUC of 0.788 and 0.775 (radiomics) AUC of 0.777 and 0.767 (combined model) AUC of 0.768 and 0.623 (clinical model) | The radiomics signature showed a robust model for identifying the MSI status |
| Pei [91] | 2022 | R | 762 | MSI status | M | 340 | AUC of 0.74 and 0.77 for the combined nomogram | The radiomics combined nomogram could be used to predict MSI status. |
| Cao [92] | 2021 | R | 502 | MSI status | M | 1037 | 32 RF showed correlation with MSI status. AUC of 0.898–0.964; ACC of 0.837–0.918; S of 0.821–1 for the combined nomogram | CT-based radiomics can predict MSI status |
| Wu [93] | 2019 | R | 102 | MSI status | M | 606 | AUC 0.961 and 0.875 for predicting MSI status | Radiomics analysis of iodine-based MD images with DECT can predict MSI status |
| Golia Pernicka [94] | 2019 | R | 198 | MSI status | M | 254 | AUC of 0.80 and 0.79 (combined model) AUC 0.74 and 0.76 (clinical and radiomics model, respectively) | Preoperative prediction of MSI status via radiomics can improve the treatment selection |
| Liu [95] | 2020 | R | 15 | LN metastasis | M | 107 | 73 RF were significant AUC 0.88 | Some RF showed significance in differentiating nonmetastatic LN from metastatic LN. |
| Cheng [96] | 2022 | R | 191 | LN metastasis | NA | NA | AUC 0.830 and 0.712 | 9 radiomic features had significant results for LN metastasis prediction |
| Huang [97] | 2016 | R | 526 | LN metastasis | M | 150 | C index 0.718 and 0.773 for radiomics signature. C index 0.763 for the prediction nomogram. | The radiomics signature combined with clinical risk factors helps in preoperative prediction of LN metastasis. |
| Eresen [98] | 2020 | R | 390 | LN metastasis | M | 146 | ACC of 0.6538–0.6282, S of 0.8387–0.8462 and Spec of 0.4713–0.4103 for the clinical model ACC of 0.8109–0.7949, S of 0.8387–0.7436 and Spec of 0.7834–0.8462 for combined model | The texture of LN provided information about the histological status of the LN |
| Li [99] | 2022 | R | 351 | Prediction LVI | M | 3095 | AUC of the combined model was 0.843 | RF combined with clinical factors had good performance in prediction of LVI |
| Ge [100] | 2021 | R | 169 | Prediction LVI | M | 396 | AUC of 0.90 for the peri-tumoral features AUC of 0.82 for the tumor features | CT-radiomics model based on the peritumoral zone improves prediction of LVI |
| Liu [101] | 2021 | R | 57 | Lung metastasis | M | 1724 | 90 RF remained unchanged in metastatic nodules | RF could be useful for investigating pulmonary nodules |
| Markich [102] | 2021 | R | 48 | Lung metastasis | NA | 64 | C-index of 0.74 for the combined model with 4 RF | RF can help to discriminate nodules at risk of local progression |
| Giannini [103] | 2020 | R | 95 | Liver metastasis | M | 22 | Acc 0.61; S 0.73; Spec 0.47 | Radiomics model can predict the likelihood of response of liver metastasis in CRC |

**Table 2.** *Cont.*

| Author | Year | Type | N | Target | ROI | RF | Results | Conclusions |
|---|---|---|---|---|---|---|---|---|
| Taghavi [104] | 2021 | *p* | 94 | Liver metastasis | NA | NA | AUC 0.60 | Radiomics models cannot predict new liver metastases of CRC |
| Staal [105] | 2021 | R | 82 | Liver metastasis | M | 56 | C-index of 0.78 | RF from the ablation zone could help in the prediction of local tumor progression |
| Liu [106] | 2022 | R | 63 | Liver metastasis | M | 851 | C-index 0.758 and 0.743 for OS<br>AUC for the 1-y survival 0.850 and 0.694<br>AUC for the 2-y survival 0.845 and 0.909<br>AUC for the 3-y survival 0.819 and 0.835 | Radiomics signature based on CT images can predict the outcome of hepatic arterial infusion chemotherapy |
| Giannini [107] | 2020 | R | 38 | Liver metastasis | M | 24 | S 0.89 and 0.90; Spec 0.85 and 0.42 for HER2 therapy response | This method is effective in predicting behavior of metastasis to HER2 treatment |
| Creasy [108] | 2021 | R | 120 | Liver metastasis | SA | 254 | 44 RF with $p < 0.05$ | There are RF that showed different distributions between patients with liver recurrence |
| Taghavi [109] | 2021 | R | 90 | Liver metastasis | M | 1593 | C Index of 0.79 in the combined model; 0.78 for the radiomics model; 0.56 for the clinical model | CT-based radiomics pre-ablation could help to predict local progression |
| Starmans [110] | 2021 | R | 76 | Liver metastasis | M | 564 | AUC 0.69 for predicting dHPG | This model has potential for automatically distinguishing dHGP from rHGP |
| Cheng [111] | 2019 | R | 126 | Liver metastasis | M | 20 | AUC of 0.926 and 0.939<br>C-index of 0.941 and 0.833 | A radiomics model can predict the HGPs of liver metastasis of CRC |
| Tharmaseelan [112] | 2022 | R | 47 | Liver metastasis | SA | 4 | Differentiate the images into 5 groups in function of the heterogeneity | RF could characterize the heterogeneity in liver metastasis of CRC |
| Devoto [113] | 2022 | R | 24 | Liver metastasis | A | NA | The metastatic liver was more heterogeneous ($p < 0.05$) | RF can differentiate a normal appearing metastatic liver from a non-metastatic liver |
| Dohan [114] | 2020 | R | 110 | Liver metastasis | M | 20 | 3 RF with $p < 0.005$ for predicting OS | RF was able to predict OS and identify good responders better than RECIST 1.1 criteria. |
| Taghavi [115] | 2021 | R | 91 | Liver metastasis | A/M | 1767 | AUC of 0.71; 0.86 and 0.86 | RF can provide valuable biomarkers to identify patients with a high risk for liver metastasis |
| Li [116] | 2022 | R | 323 | Liver metastasis | M | 1288 | AUC 0.79 and 0.72 | Combined model can provide biomarkers to identify patients with high risk of LmCRC |
| Li [117] | 2020 | R | 100 | Liver metastasis | M | 841 | AUC 0.90; 0.86; 0.906 and 0.899 | Nomogram with RF and clinical risk allows a better classification of liver metastasis |

**Table 2.** *Cont.*

| Author | Year | Type | N | Target | ROI | RF | Results | Conclusions |
|---|---|---|---|---|---|---|---|---|
| Rocca [118] | 2021 | R | 30 | Liver metastasis | M | 22 | General Acc of 0.933 | CT-based radiomics can detect LmCRC |
| Lee [119] | 2020 | R | 2019 | Liver metastasis | M | 4096 | AUC of 0.747 in prediction 5-year liver metastasis | Combined model improved the performance |
| Huang [120] | 2018 | R | 346 | Perineural invasion | M | 29 | C index 0.817 for combined nomogram | Combined nomogram was easy and effective |
| Li [121] | 2020 | R | 207 | Perineural invasion Gene expression | M | 306 | AUC of 0.793 (PNI prediction) AUC of 0.862 (KRAS prediction) | Machine learning models can predict PNI and KRAS mutation in CRC patients |
| Li [122] | 2021 | R | 303 | Perineural invasion | M | 3095 | AUC of 0.828 and 0.801 for the combined model for predicting PNI status | The combined model can help to evaluate the PNI status |
| Li [123] | 2020 | R | 779 | Peritoneal metastasis | SA | 8900 | AUC of 0.855 for combined model AUC of 0.764 and 0.771 for radiomics and clinical | Combined model, with CT-based radiomics, can be applied in the prediction of PM |

Radiomics features can also be used for the prediction of some risk factors. Hong et al. obtained an AUC of 0.799 for the combined model (radiomics-CT staging) for the prediction of high-risk colon cancer [66]. Ge et al. try to use the radiomics features for the differentiation between mucinous (MA) and non-mucinous (NMA) rectal cancer. MA rectal cancer has a poorer prognosis than NMA. They obtained an AUC of 0.93 in both training and validation cohorts [67]. Hu et al. concluded that some radiomics features are stable enough to be used for treatment monitoring and prognosis prediction [68]. Dou et al. concluded that radiomics features could help T staging [69]. Liu et al. obtained an AUC of 0.945 and 0.754 (primary and validation cohorts) in the prediction of metastatic CRC. [70]. Huang et al. obtained an AUC of 0.812 and 0.735 (training and validation cohorts) in the differentiation between low-grade and high-grade CRC [71]. Finally, Liang et al. obtained an AUC of 0.792 and 0.798 (training and validation cohorts) in the preoperative differentiation between stages I-II and III-IV in patients with CRC [72].

In terms of the use of radiomics features for predicting gene expression, Badic et al. concluded that there are genes (ABCC2, CD166, CDKNV1 and INHBB) whose expression changes are significantly associated with radiomics features [73]. Chu et al. conclude that there is an association between radiomics signature and the expression of CXCL8 [74]. Huang et al. also studied gene expression in addition to recurrence prediction. They found 10 between 1037 RF which had a significant association with recurrence prediction and with the gene expression of PECAM11, PRDM1, AIF1, IL10, ISG20 and TLR8 [75]. Hoshino et al. tried to predict the differences in tumor mutation burden (TMB) between primary and metastatic lesions using radiogenomics. They concluded that radiogenomics can infer the TMB status [76]. On the other hand, Yang et al. had significant results for the prediction of the mutations in KRAS/NRAS/BRAF genes of patients with CRC [77]. Shi et al. also studied the prediction of KRAS/NRAS/BRAF mutations in patients with CRC and liver metastasis. The AUC for the combined model (radiomics-clinical) was 0.995 and 0.79 for the primary and validation cohorts [78]. Other studies predicted KRAS status with similar results. González-Castro et al. studied the detection of KRAS mutation using radiomics based on CT images. Their highest accuracy was 0.83 [79]. Wu et al. had a very interesting study. They compared hand-crafted (HC) radiomics with deep learning (DL) based radiomics in predicting KRAS mutation. The combined model with HC-radiomics + DL-Radiomics, obtained better results [80]. He et al. used radiomics features from CT images to predict the KRAS mutation in CRC. They also used a model based on the deep learning method with a residual neural network (ResNet) for predicting KRAS status [81]. Hu et al. made a prediction of the KRAS mutation by comparing the three different phases of a CT study (non-contrast, arterial-phase, and venous-phase). They achieved the best results with radiomics features from the arterial and venous phases (AUC 0.826). The results for the combination of the three phases showed the worst AUC but the sensitivity, specificity and accuracy were higher [82]. Jang et al. compared a deep learning model and a radiogenomics model for predicting the KRAS mutation. They obtained better results with the radiomics model [83]. Xue et al. also studied the KRAS prediction. They obtained an AUC of 0.75 and 0.84 for the 2D and 3D radiomics models, respectively. They obtained better results with a combined model using clinical information (AUC of 0.92) [84]. In the prediction of BRAF status, Xue et al. obtained remarkable results. The AUC of the score with the five selected radiomics features was 0.93 and 0.87 for the training and validation cohorts. They also conducted a nomogram with clinical information and radiomics features (AUC of 0.95 and 0.88) [85]. Negreros-Osuna et al. also studied the radiomics features associated with 5-year OS. They found that some radiomics features were significantly different between BRAF mutation and BRAF wild-type [86].

In recent years, studies about the application of radiomics in the prediction of microsatellite instability have become more frequent. Fan et al. studied the association between radiomics features and microsatellite instability (MSI). They found that a radiomics model had better results than a clinical model to predict the MSI, but the combined model (clinical and radiomic) had the best results [87]. Other studies achieved similar

results. Li et al. concluded that their combined model could predict the MSI status in CRC [88]. Ying et al. established a clinical-radiomics nomogram for the preoperative prediction of MSI status in CRC patients. They achieved an AUC of 0.87 and 0.90 in the training and validation set for predicting MSI status [89]. Chen et al. studied the prediction of microsatellite instability using RF and clinical information. In their results, the radiomics signature obtained better results than the combined model [90]. Pei et al. used a radiomics nomogram with RF, clinical information, and pathological information for predicting MSI status. The radiomics nomogram with clinical information and RF obtained an AUC of 0.74 and 0.77 for the training and validation cohorts in the discrimination between high MSI (MSI-H) and no MSI-H [91]. Cao et al. found 32 radiomics features that are associated with the MSI status [92]. Wu et al. studied the application of radiomics analysis of iodine-based material decomposition images with dual-energy CT (DECT) for predicting MSI status in CRC. They obtained an AUC of 0.961 and 0.875 (training and test cohorts) predicting MSI status [93]. Finally, Golia Pernicka et al. concluded that radiomics features could help to the determination of MSI status in CRC [94].

Another application of radiomics in CRC is the determination of lymph node (LN) metastasis. Liu et al. concluded that 73 radiomics features were significant for differentiating nonmetastatic LN from metastatic LN in patients with colorectal mucinous adenocarcinoma [95]. Cheng et al. used multiphase CT for metastatic LN prediction. They obtained nine significant RF [96]. Huang et al. concluded that the radiomics signature combined with clinical risk factors facilitates the preoperative individualized prediction of LN metastasis [97]. Eresen et al. compared a clinical prediction model (based on LN size) and a combined model. The results were better for the combined model [98]. Li et al. compared a combined model with an RF-only model for the prediction of lymphovascular invasion (LVI). Their best results were with the combined model. One interesting thing in this study was that they used radiomics features from the intra-tumoral area (1490 RF) and peri-tumoral area (1605 RF) [99]. Finally, Ge et al. also concluded that the peri-tumor features can improve the prediction performance of LVI in patients with rectal cancer [100].

There are very few articles that used radiomics to investigate the metastatic pulmonary nodules in CRC. Liu et al. found that some RF remain unchanged from small to large malignant nodules, while they do not remain stable in benign nodules [101]. Markich et al. tried to assess local control of CRC lung metastases treated with radiofrequency ablation. They concluded that RF could help to discriminate nodules with a risk of local progression that could benefit from complementary treatment [102].

Radiomics have also been used in CRC patients with liver metastases. In the field of treatment response and progression prediction of liver metastasis, Giannini et al. demonstrate that the radiomics model can predict the response to chemotherapy of metastatic liver lesions [103]. Taghavi et al. in their prospective study had non-significant results for predicting new liver metastases in patients who will undergo ablation for liver metastases of CRC [104]. Staal et al. studied the prediction of local tumor progression after ablation of colorectal liver metastasis. The best results were obtained with the combined model [105]. Liu et al. studied the survival of hepatic arterial infusion chemotherapy. They also studied the 1-year, 2-year and 3-year survival [106]. There are other studies with similar results. Giannini et al. tried to predict the response of targeted HER2 treatment in liver metastasis [107]. Creasy et al. tried to predict liver recurrence, extrahepatic recurrence, or no evidence of disease after the resection of liver metastases from CRC [108]. Finally, Taghavi et al. had significant results for the progress prediction of liver metastases [109].

There are few studies that investigate the histopathological growth patterns of liver metastasis and liver heterogeneity in CRC patients. Starmans et al. had significant results for predicting a pure histopathological growth pattern (100% desmoplastic or dHGP) from a replacement histopathological pattern (rHGP) [110]. Cheng et al. also tried to predict the histopathologic growth pattern (HGP) of colorectal liver metastasis [111]. Tharmaseelan et al. studied the tumor heterogeneity in liver metastasis of CRC [112]. Finally,

Devoto et al. demonstrated with a radiomics analysis that a normal apparently metastatic liver was significantly more heterogeneous than a non-metastatic liver ($p < 0.05$) [113].

In the field of survival prediction in CRC patients with liver metastasis, Dohan et al. tried to predict OS. They found three significant RF which predict OS [114].

Finally, in the prediction of liver metastasis in CRC patients, Taghavi et al. studied the value of the RF for predicting the high risk of liver metastases in patients with CRC [115]. Li et al. had an AUC of 0.79–0.72 (in the internal and external validation cohort, respectively) in the prediction of liver metastasis in patients with CRC [116]. Li et al. also studied the performance of a nomogram for predicting liver metastasis [117]. Rocca et al. obtained a global accuracy of 0.933 for predicting liver metastasis in patients with CRC [118]. Finally, Lee et al. tried to improve the performance of a clinical model for the prediction of 5-year liver metastasis using a combined model. The AUC for the combined model was better [119].

In the evaluation of perineural invasion, there are a few studies with radiomics. Huang et al. developed a radiomics prediction model based on radiomics and CEA levels, and they obtained better results than the radiomics-only model [120]. Li et al. tried to predict the perineural invasion (PNI) and the KRAS mutation. They concluded that machine learning can predict PNI and KRAS mutation [121]. Finally, Li et al. obtained an AUC of 0.828 and 0.801 (training and validation) for the prediction of PNI [122].

The application of radiomics in the prediction of peritoneal metastasis (PM) in CRC patients has been sparsely explored. Li et al. obtained an AUC of 0.855 for the combined model. They concluded that the combined model can potentially be applied in the individual preoperative prediction of PM in CRC patients [123].

## 4. Future

*The Radiology Department of the Future*

Improving the diagnostic performance of all these models will undoubtedly be one of the main objectives of the researchers in the next years. Although major progress had been made in the last decades, there are still notable limitations: malignant lung nodules have high variability among themselves and can be located in any area of the lung; therefore, they can be hidden by the rest of the thoracic structures, such as the ribs or the diaphragm. Something common in most studies is that they report problems when detecting especially small (<3 mm) nodules or subsolid, in ground glass, and in differentiating them from some structures, such as yuxtapleural blood vessels, for example. If these systems are to be implemented, criteria should be established for how and when to use them, always considering that the scientific evidence shows some improvement in prognosis or reduction in mortality in such situations, something that we have lacked until now. At the workload level, it would be convenient to use them with caution since they could produce a paradoxical effect by increasing it. A CT image provides a wealth of information for the radiologist, and CAD, in turn, as well. The CAD could detail the most relevant aspects of the image in a complementary way, but it could also increase the time for the analysis of the images since the radiologist would have to analyze the information from both sources, especially if the lack of performance in the system could affect irreversible decisions. Most of the current diagnostic algorithms make use of advanced techniques, rather than more conventional CAD models that are being relegated. This may be a clear sign that the way forward in the future is through more complex as well as promising variants, such as Deep Learning and CNN. Their high diagnostic performance and their constant improvement through periodic reviews make it probable that, in the future, they will obtain sufficient performance to start doing larger multicenter studies that clarify whether it is a good idea to take them to hospitals. As for AI, although it tries to emulate natural intelligence, some experts believe that in the field of radiology, it will always be used as a complement to the radiologist's work, and therefore, it will never replace their work. Among the reasons for this, the inability of AI to emulate other capabilities of natural intelligence is highlighted, such as abstraction or generalization, and, above all,

that diseases sometimes have atypical presentations, something that an algorithm trained in a base with eminently typical injuries could misdiagnose [1]. Although this technology used as a complement to the radiologist's work has the ability to reduce the workload, it does not have to entail a reduction in the demand for these in hospitals. It is more feasible that this earned time is used in the rest of their duties, highlighting communication with the patient. During the healthcare process, patients demand qualities that even the most punctual technology still cannot offer today, such as empathy or communication skills. This fact is especially relevant given the path towards a more technological medicine, which should not be a step towards colder and more systematic medicine. All the work that machines and computers save us should be dedicated to patients, paradoxically making us more humane.

## 5. Discussion

When we compare the models of CAD on lung disease diagnoses, we can observe certain limitations. To begin with, some studies establish different parameters regarding the study of the nodules, so that a positive result for one could be a negative one for another. Differences in terms of the size limit, protocols in the use of obtained data, lung segmentation methods, evaluation methods, etc., represent a significant bias when comparing them. In terms of establishing actual contrasts, the current comparison between different diagnostic algorithms is complicated due to the differences in the databases and the methodology used. The availability of databases with quality information, correctly labeled, and representative of the population, is scarce. Usually, the databases with correctly labeled information are small and from isolated hospitals; therefore, they are not representative. Furthermore, the public and larger databases such as the LIDC-IDRI have data from multiple centers, but they do not have label information according to the reference standard for the diagnosis of the disease. Another important bias is related to databases since after training the model in open-access data sets, they provide results after validating or testing the model in private databases. Although this provides more reliable results, using hospital cohorts of patients who are going to have histopathological confirmation reduces the possibility of comparison between studies since the cohorts can be very different from each other. On the other hand, the use of public databases, although providing more comparable results is not without bias since the LIDC-IDRI database and its derivative LUNA16 (the most widely used) do not contain histopathological confirmation of their findings, for which the reference standard used in the studies does not correspond to that of the clinical practice (pathological anatomy), this being an obstacle to verify the real utility of these methods.

In the field of radiomics in CRC, the classic problems are still latent. There are not enough prospective studies and the methodology is also not standardized. We can see that the number of features extracted varied greatly from article to article, even if they have the same target. Many of the studies describe the features they extract, but many others simply reflect the number of them that were taken into account. Regarding these problems, we saw that there are hints of trying to change things. Initiatives such as IBSI, or studies that already include cohorts from different hospitals, are becoming more and more frequent in the scientific literature. Besides this, we cannot forget that the application of radiomics to a normal clinical way of decision-making has to be conducted in a way that the radiologist could perform it quickly and easily. All the new advances have to be for the improvement of patient management, and the improvement of the radiologist's work.

The application of CAD and radiomics to dual-energy X-ray imaging has not been adequately studied. Dual-energy chest radiography and more recently dual-energy or spectral CT have become commercially available and are routinely employed at many large medical centers. Advances in spectral CT are anticipated with the recent commercial introduction of photon counting CT. As illustrated in Figures 6 and 7 dual energy chest radiography permits one to obtain a conventional digital chest radiographic image as well as additional subtraction images with the bones removed (soft tissue image) and with

the soft tissue structures removed (bone image). With this capability, the radiologists' ability to detect subtle pulmonary nodules improved as well as their ability to discriminate between benign and malignant lesions [124,125]. It is anticipated that spectral imaging will also improve the power of CAD and Radiomics when applied to three or more distinctly different images of the same anatomical structures rather than to a single image. Dual-energy or spectral imaging compared to conventional single-energy X-ray imaging also provides reliable quantitative information that could be used to reduce false positives. It is likely that Radiomics will also benefit from the accurate and reproducible quantitative input that spectral x-ray imaging provides.

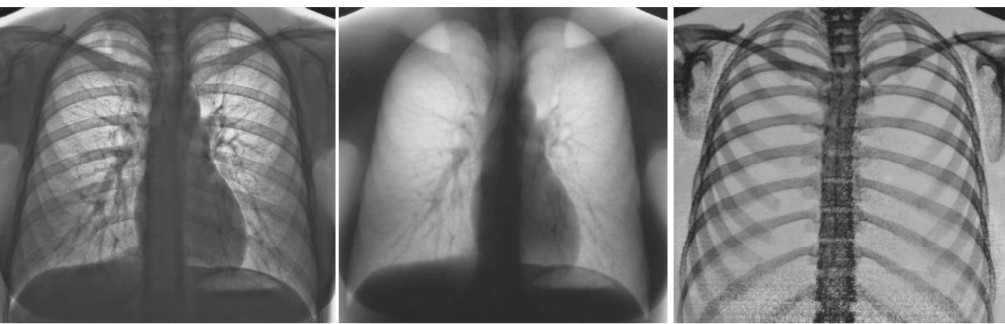

**Figure 6.** Images obtained on a prototype dual energy digital chest radiographic unit at the University of Alabama Birmingham circa 1985 with a single exposure: the conventional digital image was obtained by adding the low and high energy images (**left**); soft tissue image (**center**) and bone image (**right**) [124,125].

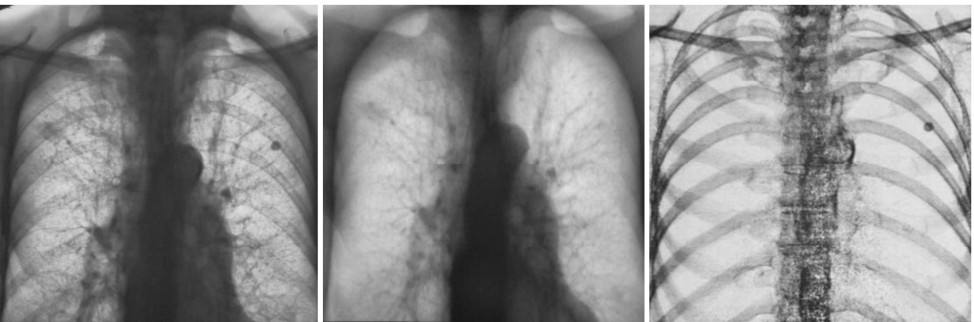

**Figure 7.** Images obtained on a prototype dual energy digital chest radiographic unit at the University of Alabama Birmingham circa 1985 with a single exposure of a patient with lesions present in both sides of the lung in the conventional digital image (**left**); Soft tissue (**center**) and bone (**right**) images of the patient. The nodule in the lung left side is calcified and benign. The nodule on right side is present in the soft tissue image and not in the bone image, was not calcified and was cancer [124,125].

## 6. Conclusions

Radiology is one of the medical specialties that has experienced great changes in the last 50 years. The discovery of CT and MRI, and the digitization at the end of the twentieth century, made radiology go hand in hand with the technological evolution of society. The high technological developments experienced in recent decades brought AI systems. These systems were quickly applied to the health field as well as to other economic sectors. In the beginning, AI (CAD) applied to medicine emerged as a possible substitute for radiologists. The possibilities for AI that experts glimpsed were endless. The initial efforts were mainly focused on the diagnostic capacity of AI systems. With the development of the first applications of AI, it was discovered that replacing the work of the clinician was not easy. Some of the AI systems developed were able to answer a specific question. However, many of the medical decisions made day-to-day require reflection and reasoning that is not capable of being imitated even with the most sophisticated AI techniques.

In the field of medical imaging, this initial outburst gave way to first-generation CAD systems, which attempted to replace radiologists. This initial trivialization of the radiological profession made little headway. It was seen that many of the AI applications developed were not cost-effective due to the complexity and high specialization required to make decisions based on medical images. The radiologist faces a wide range of pathologies in their routine. Therefore, the development of AI in the radiological field should provide more than the ability to give a specific diagnosis.

Gradually, with increased cooperation between developers and clinical staff, AI focused on improving the diagnostic process itself, rather than providing a final diagnosis. In this sense, in the field of radiology, great advances were made in CT and MRI. It was along these lines that second-generation CAD systems emerged. These systems no longer aspire to replace radiologists. They provide a second opinion in complex situations or when AI presents a significant advantage. In this sense, these systems seek to liberate the radiologist from the most repetitive tasks, as well as improve other phases of the diagnostic process (acquisition, processing and post-processing of images).

In recent years, there has been a new boom in certain AI applications. In the congresses of the different radiology societies, the main topic is the use of AI to improve the day-to-day practice of radiologists. This renewed enthusiasm is due to the development of applications, such as radiomics. Radiomics has exploded with force because of the desire to provide increasingly personalized medicine. The reality is that today radiomics is far from being applied in a simple and intuitive way to the daily diagnostic course as we know it. The complexity of the algorithms and the aspects of the process remain unknown even to experts in the field, meaning that the use we will probably see in the coming years continues to be that of a "support tool". Although the idea of being able to replace a biopsy with a simple click is very tempting, for the moment caution should reign. Radiomics should be a guide tool for the reference techniques already proven. In this sense, what should be sought in the coming years is not to replace other techniques but to think about how to complement them to be more precise and efficient.

It has been more than 30 years since the first CAD researchers predicted the end of many of the medical professions as we know them. Being an eminently technological specialty, radiology was the focus of these prophecies. However, we could see in these years that radiologists were not affected by technological advances. They were at the forefront of this evolution and demonstrated increased levels of patient care.

Thanks to the collaboration established between developers and radiologists, the future is bright. As seen in the history of AI, this path will continue with ups and downs in the coming years. At present we are again in a time of change. However, unlike 30 years ago, we see that the goal of AI applications has changed. Before it was sought to replace the radiologist, currently it is attempting to provide the radiological profession with a greater level of precision and patient care. News about the end of the radiologist profession is also becoming less frequent, reflecting researchers' growing knowledge of the complexity of radiology practice.

In short, it seems that in the future AI will change medical practice as we know it. In the field of radiology, we will see fundamental improvements in diagnostic capabilities thanks to the increasingly sophisticated tools that will become available. However, AI will not replace the intuition, consensus and decision-making capacity of human reasoning. While it is true that decision algorithms will continue to be developed in certain tasks, there will continue to be an important limitation when it comes to resolving conflicts derived from different points of view. In an increasingly multidisciplinary medical practice, this handicap is an important limitation to the initial idea researchers had of replacing the human component in medical decision-making.

**Author Contributions:** Conceptualization, J.P.-Á., G.T.B., A.V. and M.S.-B.; methodology, J.P.-Á., G.T.B., A.V. and M.S.-B.; writing-original draft preparation, J.P.-Á., G.T.B., A.V., R.G.-F. and M.S.-B.; writing-review and editing, G.T.B., R.G.-F., S.B.-G., E.H.Z. and M.S.-B.; supervision, G.T.B., R.G.-F., E.H.Z., S.B.-G and M.S.-B. All authors have read and agreed to the published version of the manuscript.

**Funding:** This research received no external funding.

**Data Availability Statement:** No new data were created or analyzed in this study. Data sharing is not applicable to this article.

**Conflicts of Interest:** The authors declare no conflict of interest.

## Abbreviations

Acc: Accuracy; AI: Artificial Intelligence; AUC: Area Under the Curve; BAcc: Balanced Accuracy; C-Index: Concordance Index; CAD: Computer-Aided Diagnoses; CE: Contrast Enhanced; CEA: Carcinoembryonic Antigen; CNN: Convolutional Neural Network; CR: Complete Response; CRC: Colorectal Cancer; CT: Computer Tomography; DBN: Deep Belief Network; DECT: Dual Energy Computer Tomography; DL: Deep Learning; DL-CAD: Deep Learning based Computer-Aided Diagnoses; EGFR: Epidermal Growth Factor Receptor; ExHOG: Extended Histogram of Oriented Gradients; FPSO: Fuzzy Particle Swarm Optimization; FS: Free Survival; GR: Good Response; HC: Hand Crafted; HER2: Human Epidermal Growth Factor Receptor 2; HGP: Histopathological Grow Pattern; HR: Hazard Ratio; IBSI: Imaging Biomarker Standardization Initiative; LARC: Locally Advanced Rectal Cancer; LASSO: Least Absolute Shrinkage and Selection Operator; LDCT: Low Dose Computer Tomography; LIDC-IDRI: Lung Image Database Consortium-Image Database Resource Initiative; LmCRC: Liver Metastases of Colorectal Cancer; LN: Lymph Nodes; LVI: Lymphovascular Invasion; M: Manual; MA: Mucinous rectal cancer; MD: Material Decomposition; MIP: Maximum Intensity Projections; ML: Machine Learning; MO-FS: Multi-Objective based Feature Selection; MRI: Magnetic Resonance Imaging; MSI: Microsatellite Instability; N: Number of patients; NA: Not Available; NAR: Neoadjuvant Rectal score; NCE: Non-Contrast Enhanced; NMA: Non-Mucinous rectal cancer; NPV: Negative Predictive Value; OR: Odds Ratio; OS: Overall Survival; P: Prospective; PACS: Picture Archiving Communication Systems; PET: Positron Emission Tomography; PFS: Progression Free Survival; PM: Peritoneal Metastasis; PNI: Perineural Invasion; PPV: Positive Predictive Value; R: Retrospective; ResNet: Residual Network; RF: Radiomics Features; ROI: Regions Of Interest; RPN: Region Proposal network; rs: Spearman's rank correlation coefficient; S: Sensitivity; SA: Semi-Automatic; SE: Squeeze Excitation; Spec: Specificity; SVM: Support Vector Machine; TBS: Tumor Burden Score; TMB: Tumor Mutation Burden; WLTB: Whole Liver Tumor Burden.

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
