# Peer review of "Digital Medical X-ray Imaging, CAD in Lung Cancer and Radiomics in Colorectal Cancer: Past, Present and Future"

_applsci, doi:10.3390/app13042218_

Round 1

Reviewer 1 Report

Well written with clear conclusion. 

Author Response

Dear reviewer, we appreciate your comments. 

Respectfully, 

The Authors.

Reviewer 2 Report

This paper states the status and development on the digital medical X-Ray imaging, CAD and radiomics, from the past, present and future, gives relevant analysis, including methods and algorithms and applications. The analysis of the paper is more detailed, there are relevant case analysis, however, this paper is not very innovative, I don't see the author's own new ideas and discoveries, the author needs to add some content. The details are as follows:

(1) What are the main problems and shortcomings of digital medical X-Ray imaging, CAD and radiomics? What are the key technologies to break through?

(2) At present, with the continuous development of deep learning, neural network, integrated learning and other technologies, and so on, can we solve the technical problems of medical X-Ray imaging? What are the outstanding new ideas?

(3) The third part states a lot of content, most of which is the thought of existing literature. What is the author's core idea? What are the new developments? The author does not give a conclusive summary and description.

(4) in part 4, The author mentions that the application of CAD and radiomics to dual energy x-ray imaging has not been adequately studied, What does the author think of the future? What new technology should be adopted?

(5) This review paper does not well highlight the application and development of technologies and algorithms. What is the real significance for Digital Medical X-Ray Imaging, CAD and Radiomics? It is suggested that the author give his own contribution and innovation of the paper in the first part, and give own technical views in the other parts.

Reviewer 3 Report

In this paper, the authors proposed a review paper to point out the key aspects of the digital medical imaging revolution, to review its current status, to discuss its clinical translation in two major areas: lung and colorectal cancer.

The paper’s subject is very interesting and useful for researchers who do investigations in the same topic. Therefore, I highly recommend this paper for publication in this journal but before that, I have some comments on the text that should be addressed before publication:

Comments:

1) The objectives of this article are to point out the key aspects of the digital medical imaging revolution to discuss  its clinical translation in two major areas: lung and colorectal cancer. But title of this review paper is general. It doesn’t address specifically application of medical imaging for lung and colorectal cancer. I would suggest to make the title more specific.

2) The sentence in line 19 “Besides, it can also focus on aspects of the image that suggest pathology such as computer-aided diagnosis (CAD)” is not clear to me. Can author explain it a bit more?

3)Figure 1 quality is not good. If possible, please replace it with a higher quality one.

4) It seems figure 1 has been copied from another reference. If so, it would be nice if the reference of source paper is added in the caption.

5)The paper is really well organized and I’m pretty happy with that

Reviewer 4 Report

The main goals of this article are to highlight the salient features of the digital medical imaging revolution, to examine its present state, to describe its clinical application in two significant fields, namely colorectal and lung cancer, and to propose future directions and problems for these methods.

The necessary citations are missing in most of the sections and subsections including Sectino2, 2.3, 3. For eg .....................Zhang C. et al propose a DL algorithm that performs the detection and classification processes in a unified way in contrast to most studies, which apply a sequential algorithm, since this reduces the possibility of accumulating errors............... . This is only an example. There are plenty of missing citations in this article.

There are very poor resolution for a few of the Figures eg Fig 1, Fig 2 and Fig 3. These figures has to be regenerated.

The author had included a good comparison of reviews in tabular for method. But, if they use graphical representation too, it will be more attractive and comparable in a readers point of view.

Round 2

Reviewer 2 Report

This revised manuscript does not directly answer the questions raised. It does not focus on the core status of the research, but it basically meets the requirements on the whole.

Author Response

Dear Revisor. 

Thank you very much for everything. We appreciate your suggestions and will take them into account for future articles.

Kind regards,

The authors.

Reviewer 3 Report

paper is ready for publication in the present form

Author Response

Dear reviwer.

Thank you very much for everything.

Kind regards,

The authors.

Reviewer 4 Report

Thank you for including my suggestions in the paper. However, I feel your manuscript still needs some minor corrections before the publication.

Necessary citations are missing in Section 1, Subsection 2.1, 2.3, 3.1. Also, the statements in the subsubsections 3.2.2, 3.2.3, 3.2.5 are to be supported with necessary citations from reliable sources. Eg line number 426, 427.

Please do not highlight the text within the figure rather only need to highlight its caption.

In figure 5a and 5b, the bar graphs should be prepared using scientific tools and it is necessary to include the labels for x and y axis.

There are same colors given for different fields in Figure 5(c). (eg 2,3 and 17). Kindly provide its reason.  

Necessary citations are missing in the first and second paragraph of section 4.

There are same figure number (Figure 5) for two figures; one in page number 23 and the other in page number 25. This  is to be corrected. Besides, this necessary citations should be given to the figures taken from other sources.

The author had included necessary references in the manuscript. But, there are a lot of citation missings in it. The author has to include all the missing citations. As it is a study, the citations are playing vital role in it.
